# Morphological and Electrochemical Properties of ZnMn_2_O_4_ Nanopowders and Their Aggregated Microspheres Prepared by Simple Spray Drying Process

**DOI:** 10.3390/nano12040680

**Published:** 2022-02-18

**Authors:** Gi Dae Park, Yun Chan Kang, Jung Sang Cho

**Affiliations:** 1Department of Advanced Materials Engineering, Chungbuk National University, Cheongju 361-763, Korea; gdpark@cbnu.ac.kr; 2Department of Materials Science and Engineering, Korea University, Anam-Dong, Seongbuk-Gu, Seoul 136-713, Korea; 3Department of Engineering Chemistry, Chungbuk National University, Cheongju 361-763, Korea

**Keywords:** nanopowders, spray drying, microsphere, lithium-ion batteries

## Abstract

Phase-pure ZnMn_2_O_4_ nanopowders and their aggregated microsphere powders for use as anode material in lithium-ion batteries were obtained by a simple spray drying process using zinc and manganese salts as precursors, followed by citric acid post-annealing at different temperatures. X-ray diffraction (XRD) analysis indicated that phase-pure ZnMn_2_O_4_ powders were obtained even at a low post-annealing temperature of 400 °C. The post-annealed powders were transformed into nanopowders by simple milling process, using agate mortar. The mean particle sizes of the ZnMn_2_O_4_ powders post-treated at 600 and 800 °C were found to be 43 and 85 nm, respectively, as determined by TEM observation. To provide practical utilization, the nanopowders were transformed into aggregated microspheres consisting of ZnMn_2_O_4_ nanoparticles by a second spray drying process. Based on the systematic analysis, the optimum post-annealing temperature required to obtain ZnMn_2_O_4_ nanopowders with high capacity and good cycle performance was found to be 800 °C. Moreover, aggregated ZnMn_2_O_4_ microsphere showed improved cycle stability. The discharge capacities of the aggregated microsphere consisting of ZnMn_2_O_4_ nanoparticles post-treated at 800 °C were 1235, 821, and 687 mA h g^−1^ for the 1st, 2nd, and 100th cycles at a high current density of 2.0 A g^−1^, respectively. The capacity retention measured after the second cycle was 84%.

## 1. Introduction

Transition metal oxides (TMO) with high theoretical capacities have been actively researched as prospective anode materials for lithium ion batteries (LIBs) [1,2,3,4,5,6,7,8,9,10,11,12,13,14,15,16,17,18,19,20,21]. In particular, multicomponent transition metal oxide nanostructures are considered to exhibit good cycling and rate performances because of their structural stability during cycling, high surface-to-volume ratio, and short diffusion length for Li ions [1,2,3,4,5,6,7,8,9,10,11,12,13,14,15].

ZnMn_2_O_4_ is one of the most attractive binary transition-metal oxides because of its low oxidation potential and low material cost [14,15,16,17,18,19,20,21]. Lately, several studies have reported the electrochemical properties of ZnMn_2_O_4_ powders with various structures, prepared by liquid solution methods [10,14,15,16,17,18,19,20,21]. For instance, Deng et al. reported the synthesis of agglomerated ZnMn_2_O_4_ nanoparticles by a single-source precursor route, which exhibited high rate capability with a capacity of 330 mA h g^−1^, even after 35 cycles at 600 mA g^−1^ [18]. Similarly, Courtel et al. prepared ZnMn_2_O_4_ nanopowders by a simple and easily scalable co-precipitation route. According to their study, ZnMn_2_O_4_ nanopowders sintered at 800 °C (<150 nm) gave the best battery performance, exhibiting a capacity of 690 mA h g^−1^ at a low current density of 0.1 C [14]. Likewise, another study demonstrated that ZnMn_2_O_4_ nanopowders with a diameter of 5 nm prepared by a hydrothermal synthesis gave a capacity of 430 mA h g^−1^ when cycled 100 times at 0.1 C [15]. Luo et al. synthesized mesoporous ZnMn_2_O_4_ microtubules via a facile biomorphic strategy employing biotemplate [16]. The 1D-mesoporous structure of ZnMn_2_O_4_ microtyblues facilitated superior electrochemical properties of high capacity and rate capability.

Spray drying is considered as commercial process to obtain dried powders. In general, liquid or colloidal solution was applied to spray drying process resulting in spherical dried powders. Thus far, a number of literatures have been reported to synthesis of electrode materials for lithium ion batteries [22,23,24]. Hou et al. reported synthesis of LiNi_0_._815_Co_0_._15_Al_0_._035_O_2_ cathode materials by spray drying and a high-temperature calcination. The metal salt dissolved solution was applied to preparation of precursor powders [25]. Asenbauer et al. synthesized microsized, nanocrystalline Zn_0_._9_Fe_0_._1_O-C secondary particles by three spray-drying steps. The large secondary particle size of about 10–15 mm facilitated handling and processing during preparation slurry and provided good electrochemical properties [26]. However, preparation of multicomponent oxide nanopowders and their aggregated powders by spray drying using aqueous spray solution should be researched due to the poor drying characteristics of metal salts under a humid atmosphere.

Herein, we have made an attempt to synthesize ZnMn_2_O_4_ nanopowders by a simple spray drying process, using mixed metal salt powders of zinc and manganese as precursors and citric acid followed by post-annealing at different temperatures. In addition, citric acid facilitates the formation of mixed metal salt powders with hollow and porous structure under a highly humid atmospheric condition. The as-prepared powders were post-annealed at temperatures between 400 and 1000 °C to form phase-pure spinel ZnMn_2_O_4_ nanopowders. However, nanopowders tend to agglomerate easily in terms of battery materials, which interferes with stable electrochemical properties. In this regard, in this study, to provide practical utilization, the nanopowders were transformed into aggregated microsphere consisting of ZnMn_2_O_4_ nanoparticles by second spray drying process. Furthermore, we have also analyzed the effect of post-annealing temperature on the electrochemical properties of the ZnMn_2_O_4_ nanopowders and their aggregated microsphere consisting of ZnMn_2_O_4_ nanoparticles, which were prepared at optimized post-treatment temperature, as an anode material for LIBs was investigated.

## 2. Materials and Methods

In this study, ZnMn_2_O_4_ nanopowders with hollow structure were obtained by applying a spray drying process followed by post-treatment at different temperatures. The precursor spray solutions were prepared by dissolving zinc nitrate hexahydrate [Zn(NO_3_)_2_·6H_2_O, 98%, SAMCHUN, Pyeongtaek, Korea] and manganese nitrate hexahydrate [Mn(NO_3_)_2_·6H_2_O, 98%, SAMCHUN, Pyeongtaek, Korea] in distilled water. The concentrations of zinc and manganese components were fixed at 0.17 and 0.33 M, respectively. Similarly, the concentration of the chelating agent citric acid (99.5%, SAMCHUN, Pyeongtaek, Korea) was kept at 0.5 M. The inlet and outlet temperatures during spray drying were controlled as 350 °C and 150 °C, respectively. The atomizer was applied by a two-fluid. The flow rate and nozzle inlet diameter (where solution is supplied into hot chamber) were 1 mL s^−1^ and 1.0 mm, respectively. Atomization pressure was 0.3 bar. The ZnMn_2_O_4_ nanopowders were obtained by spray drying the precursor solution, which were subsequently post-annealed in air at temperatures between 400 and 1000 °C for 3 h. To obtain uniformly dispersed state of colloidal solution consisting of ZnMn_2_O_4_ nanopowders, which were obtained at 800 °C post-treatment temperature, ultrasonication for 1 h was conducted. Subsequently, aggregated ZnMn_2_O_4_ microsphere powders were produced by second spray drying process. During the second spray drying, the temperatures of inlet and outlet were 300 and 120 °C, respectively. Atomization pressure was a higher value (2 bar) compared to the first spray drying. Appendix A shows a schematic diagram of the spray drying system and formation of ZnMn_2_O_4_ aggregated microsphere by a 2-step spray drying process. Additional details on characterization and applied electrochemical measurements upon samples are provided in the Appendix A.

## 3. Results and Discussion

Figure 1 shows the XRD (X’Pert PRO MPD, PANalytical, Malvern, UK) pattern of the as-prepared and post-treated powders annealed at different temperatures and standard PDF card of ZnMn_2_O_4_ [27]. The XRD pattern of the as-prepared powders, directly obtained from the spray drying process, had amorphous structure without any crystalline peaks. This implies that decomposition of metal chelates formed from the precursor metal ions and citric acid did not occur during the spray drying process. On the other hand, the XRD pattern of the ZnMn_2_O_4_ powders post-annealed at 400 °C exhibited broad diffraction peaks with low intensity. However, sharpness and intensity of the diffraction peaks increased with increase in the post-annealing temperature from 400 to 1000 °C. The mean crystallite size of the ZnMn_2_O_4_ powders post-annealed at 400, 600, 800, and 1000 °C were estimated to be 10, 20, 32, and 42 nm, respectively. It was calculated by Scherrer’s equation from the half-width of the (211) peak.

Figure 2 displays the morphology of the as-prepared and post-treated powders at temperatures of 400, 600, and 800 °C. The morphology of the as-prepared powders indicates the formation of large particles of size several tens of micrometers with a hollow morphology. We believe that the fast drying of the droplets during the spray drying process would have resulted in the generation of hollow particles. The SEM (S-4800, Hitachi, Japan) images of the post-annealed samples reveal that the spherical and hollow shape of the as-prepared particles was kept even after the post-treatment at temperatures between 400 and 800 °C. During the post-annealing conditions, the decomposition and crystallization of the as-prepared powders improved its porous structure. The post-annealed powders were transformed into ZnMn_2_O_4_ nanopowders by a simple hand-milling process, using agate mortar and pestle.

Figure 3 and Figure 4 show the low- and high-resolution TEM (JEM-2100F, JEOL, Japan) images of the ZnMn_2_O_4_ powders post-annealed at 600 and 800 °C. The ZnMn_2_O_4_ powders post-treated at 600 °C (Figure 3a,b) were composed of aggregated particles, characteristic of nanosized powders prepared by simple milling. However, the ZnMn_2_O_4_ powders post-annealed at 800 °C (Figure 4a,b) did not have aggregated particles. The high-resolution (HR) TEM images of the ZnMn_2_O_4_ nanopowders post-annealed at 600 and 800 °C (Figure 3c and Figure 4c, respectively) indicated that the particles were single crystals. The HR TEM image revealed lattice fringes divided by 0.49 nm, corresponding to (101) plane of the ZnMn_2_O_4_ [21]. The mean particle size of the powders post-annealed at 600 and 800 °C, as measured from the low-resolution TEM images, were estimated to be 43 and 85 nm, respectively. The corresponding dot-mapping images (Figure 3d and Figure 4d) indicated that the zinc, manganese, and oxygen components were uniformly distributed all over the ZnMn_2_O_4_ nanopowders.

Figure 5 shows the morphology of the ZnMn_2_O_4_ powders post-annealed at a high temperature of 1000 °C, before and after simple milling using agate mortar. The SEM image of the ZnMn_2_O_4_ powders post-annealed at 1000 °C (shown in the inset in Figure 5a) revealed that the particles were hardly aggregated. On the other hand, the milled powders (Figure 5b) had particles of submicron size with broad size distribution.

Figure 6 exhibits the morphologies of the aggregated microsphere consisting of ZnMn_2_O_4_ nanoparticles, which were prepared at 800 °C post-treatment temperature, formed by the two-step spray drying process. The nanoparticles consisting of aggregated microsphere were prepared by facile hand-milling process using an agate mortar as shown in Figure 4a,b. The colloidal spray solution containing ZnMn_2_O_4_ nanoparticles was applied in the second step of spray drying. In particular, the aggregated microsphere showed non-aggregated state between microsphere and spherical morphology, which could be advantageous for electrode materials. The crystal structure of the aggregated ZnMn_2_O_4_ microsphere powders was kept even after second spray drying process as shown in Appendix A. The surface area and pore size distribution of the aggregated microsphere powders formed by the second step spray drying process are exhibited in Appendix A. The aggregated ZnMn_2_O_4_ microsphere powders had well-developed mesopores. The BET surface areas of the aggregated microsphere powders were 15.8 m^2^ g^−1^, respectively.

Figure 7 displays the electrochemical properties of the ZnMn_2_O_4_ powders post-annealed at diverse temperatures and aggregated microsphere consisting of ZnMn_2_O_4_ nanoparticles prepared at 800 °C (denoted as 800 °C_AM). Figure 7a represents the initial potential curves of the ZnMn_2_O_4_ powders at a high constant current density of 2.0 A g^−1^. The charge and discharge curves of the powders had similar shapes, irrespective of the post-annealing temperature, owing to their same crystal structure. In the initial discharge curves, all the ZnMn_2_O_4_ powders showed one plateau at about 0.3 V, which could be attributed to the reduction of ZnMn_2_O_4_ into LiZn, metallic Zn, Mn, MnO, and Li_2_O, as confirmed by previous literatures [28,29,30,31]. The initial discharge capacities of the ZnMn_2_O_4_ powders post-annealed at 400, 600, 800, and 1000 °C and 800 °C_AM were 1530, 1083, 1236, 1050, and 1235 mA h g^−1^, respectively, and the charge capacities were 847, 757, 758, 474, and 823 mA h g^−1^, respectively. The ZnMn_2_O_4_ powders post-annealed at 400 °C had high initial discharge and charge capacities because of their fine crystallite size. The Coulombic efficiencies of the ZnMn_2_O_4_ nanopowders post-annealed at 400, 800 °C, and 800 °C_AM in the first cycle were 55, 61, and 66%, respectively. In the initial cycle, ZnMn_2_O_4_ electrodes showed commonly irreversible capacity loss, which was owing to the generation of a solid electrolyte interphase (SEI) film [28,29,30,31]. Figure 7b shows the cyclic voltammograms (CVs) of the 800 °C_AM for the first three cycles in the voltage range 0.01–3 V at a scan rate of 0.07 mV s^−1^. An intensive reduction peak at about 0.2 V was observed in the first cycle, which could be ascribed to the irreversible reduction of ZnMn_2_O_4_ with concomitant crystal structure destruction to form metallic nanograins (Zn^0^, Mn^0^) dispersed in an amorphous Li_2_O matrix and some of LiZn alloys and MnO phases [31,32]. Intensive reduction peaks at about 0.5 V were observed in the second and third cycles. Recently, the various ZnMn_2_O_4_ anode materials were reported due to their excellent electrochemical properties and their electrochemical reaction mechanism was specifically investigated by in-situ or ex-situ XRD and XANES analysis. By specific analysis, it was confirmed that the final discharge product was a mixture of Zn^0^, LiZn alloy, and MnO with a fraction of Mn^0^ embedded in Li_2_O. After second cycle, the reversible reaction is as follows: LiZn + Zn + Mn + MnO + (4 + x)Li_2_O ↔ 2ZnO + 2MnO + 9Li + xLi_2_O [31,32]. The reduction/oxidation peaks in the CV tests from the second cycle onwards overlapped very well, revealing that the ZnMn_2_O_4_ 800 °C_AM prepared by spray drying process showed good cyclability. Figure 7c displays the cycling performance of the ZnMn_2_O_4_ powders post-annealed at different temperatures and aggregated microsphere (800 °C_AM) at a constant current density of 2.0 A g^−1^. The discharge capacities of the ZnMn_2_O_4_ powders post-annealed at a low temperature of 400 °C strictly decreased with an increase in the cycle numbers. However, the ZnMn_2_O_4_ powders post-treated at high temperatures of 800 and 1000 °C showed relatively good cycling performance even at a high current density of 2.0 A g^−1^. On the other hand, 800 °C_AM electrode exhibited improved cyclic stability due to their mesoporous structure and non-aggregated state between the microspheres [33,34,35]. After 100 cycles, the discharge capacities of the ZnMn_2_O_4_ powders post-treated at 400, 600, 800, 1000 °C, and ZnMn_2_O_4_ 800 °C_AM were 431, 447, 532, 400, and 687 mA h g^−1^, respectively. In addition, after the first cycle their corresponding capacity retentions were 49, 62, 69, 71, and 84%, respectively. Results indicate that the optimum post-annealing temperature required to obtain ZnMn_2_O_4_ nanopowders with a high capacity and good cycling performance is 800 °C. and their aggregated microsphere exhibited enhanced cycling performance. Figure 7d displays the rate performance of the ZnMn_2_O_4_ 800 °C_AM. The rate performance was conducted by increasing current density from 1.0 to 4.0 A g^−1^ and finally recovered to 1.0 A g^−1^. Then, ZnMn_2_O_4_ nanopowders exhibited discharge capacities of from 880 to 594 mA h g^−1^ as increasing current density from 1.0 to 4.0 A g^−1^. In spite of the cycling at high current densities, the discharge capacity recovered to 829 mA h g^−1^ when the current density returned to 1.0 A g^−1^. Appendix A exhibited the charge and discharge profiles of ZnMn_2_O_4_ 800 °C_AM at different current densities. Despite the increase in current density, potential profile shape was well maintained, indicating their outstanding rate capability. The mesoporous structure consisting of nanoparticles contributed to an excellent rate capability. The electrochemical properties with other ZnMn_2_O_4_ materials as anode materials for LIBs reported in the previous literatures are exhibited in Appendix A. The structural advantages of ZnMn_2_O_4_ 800 °C_AM facilitated good cycle life with high reversible capacities, even compared to those of other ZnMn_2_O_4_ materials.

Electrochemical impedance spectra (EIS) of ZnMn_2_O_4_ powders post-treated at 800 °C temperatures and aggregated microspheres consisting of ZnMn_2_O_4_ nanoparticles (800 °C_AM) obtained before and after 100 cycles are displayed in Appendix A. Before cycle and after 100 cycles, the ZnMn_2_O_4_ nanopowders post-treated at 800 °C and exhibited aggregated microspheres consisting of ZnMn_2_O_4_ nanoparticles (800 °C_AM) showed similar R_ct_ (charge-transfer resistance) values. On the other hand, before cycle, ZnMn_2_O_4_ nanopowders post-treated at 800 °C temperatures showed a low slope (s, Warburg impedance coefficient) of the real part of the impedance spectra (Z_re_) versus w^−1/2^, indicating a high lithium-ion diffusion rate of nanopowders compared to aggregated microsphere. However, after 100 cycles, the slopes of two electrodes showed similar values, revealing that the ZnMn_2_O_4_ nanopowder became more agglomerated as the cycle progressed. This was confirmed by the SEM images of 100 cycles as displayed in Appendix A. The ZnMn_2_O_4_ nanopowder showed a severely aggregated state after 100 cycles. On the other hand, ZnMn_2_O_4_ 800 °C_AM microspheres maintained their spherical and porous structure even after repeated cycles.

## 4. Conclusions

In summary, ZnMn_2_O_4_ nanopowders and their aggregated microspheres were prepared by simple spray drying using mixed metal salt powders of zinc and manganese as precursors and citric acid as the chelating agent, followed by post-annealing at temperatures between 400 and 1000 °C. The phase-pure ZnMn_2_O_4_ powders with porous particles obtained by post-annealing treatment were converted into nanopowders by a simple milling process. Subsequently, the nanopowders were transformed into aggregated microspheres consisting of ZnMn_2_O_4_ nanoparticles by a second spray drying process to provide structural stability and practical utilization. Electrochemical studies reveal that the ZnMn_2_O_4_ powders post-annealed at 400 °C had a high initial discharge and charge capacities and showed poor cycling performances because of their fine crystallite size. On the other hand, aggregated microsphere consisting of ZnMn_2_O_4_ powders, which were post-annealed at 800 °C resulting in a fine size and high crystallinity, exhibiting great cycling stability with high capacities and rate capability.

## Figures and Tables

**Figure 1 nanomaterials-12-00680-f001:**
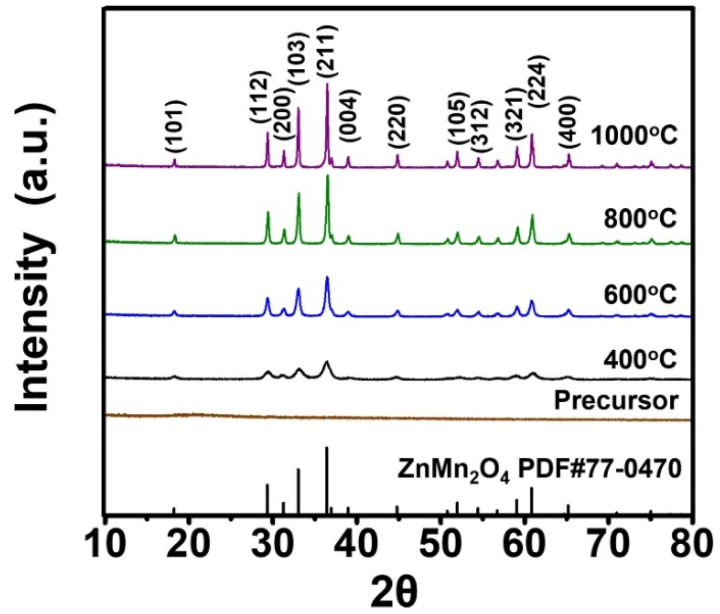
XRD patterns of the precursor and ZnMn_2_O_4_ powders post-treated at various temperatures.

**Figure 2 nanomaterials-12-00680-f002:**
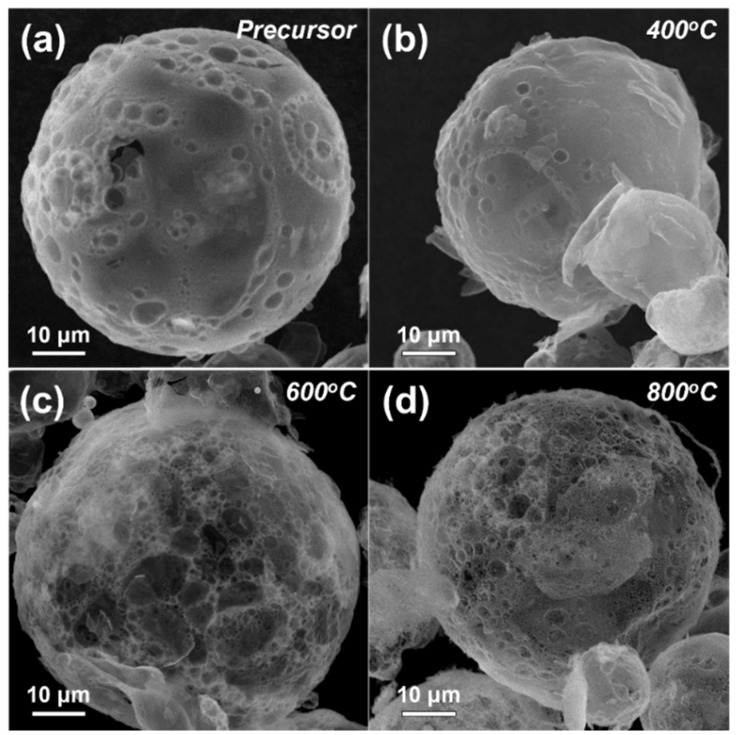
Morphologies of the precursor and ZnMn_2_O_4_ powders post-treated at various temperatures; (**a**) precursor, (**b**) 400 °C, (**c**) 600 °C, (**d**) 800 °C.

**Figure 3 nanomaterials-12-00680-f003:**
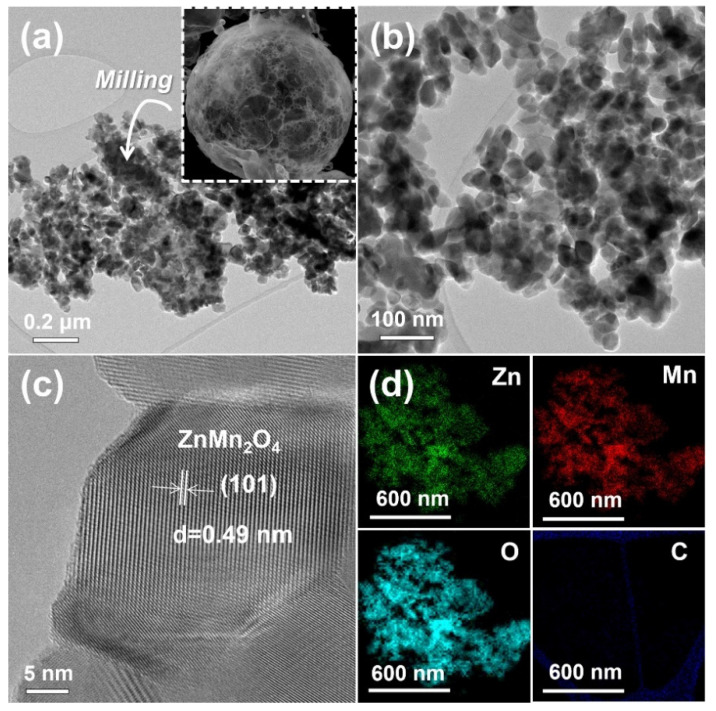
TEM and dot-mapping images of the ZnMn_2_O_4_ powders post-treated at 600 °C after simple milling process by hand using an agate mortar: (**a**) TEM image (inset SEM image of before simple milling process), (**b**) TEM image, (**c**) HR-TEM image, and (**d**) dot-mapping images.

**Figure 4 nanomaterials-12-00680-f004:**
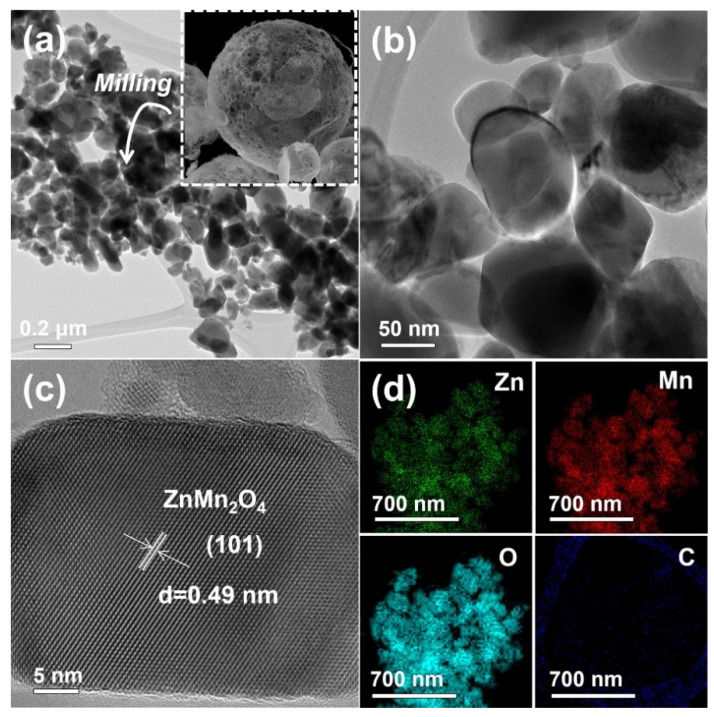
TEM and dot-mapping images of the ZnMn_2_O_4_ powders post-treated at 800 °C after simple milling process by hand using an agate mortar: (**a**) TEM image (inset SEM image of before simple milling process), (**b**) TEM image, (**c**) HR-TEM image, and (**d**) dot-mapping images.

**Figure 5 nanomaterials-12-00680-f005:**
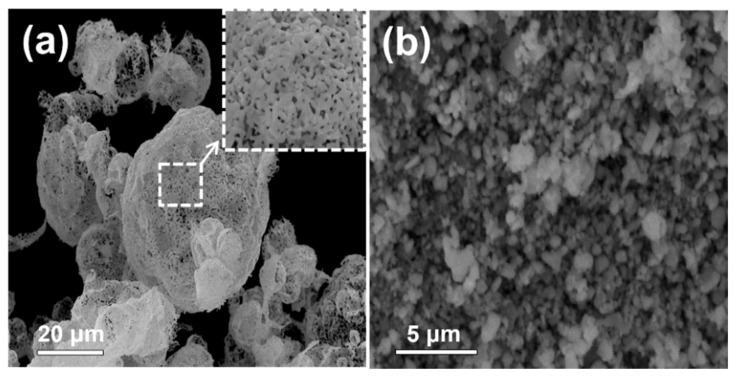
SEM images of the ZnMn_2_O_4_ powders post-treated at 1000 °C (**a**) before and (**b**) after simple milling process by hand using an agate mortar.

**Figure 6 nanomaterials-12-00680-f006:**
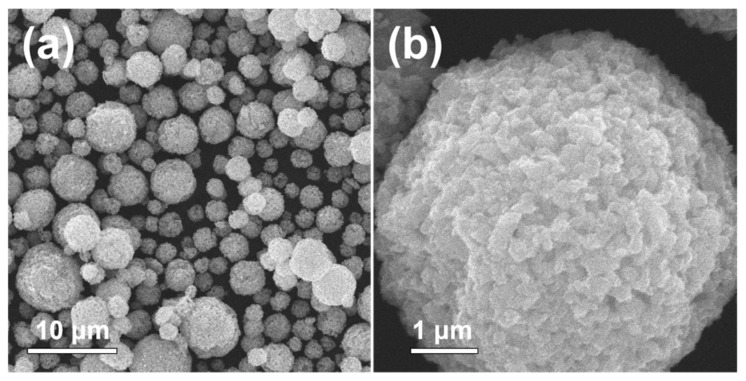
SEM images of the aggregated microsphere consisting of ZnMn_2_O_4_ nanoparticles, which were prepared at 800 °C post-treatment temperature: (**a**) low magnification image and (**b**) high magnification image.

**Figure 7 nanomaterials-12-00680-f007:**
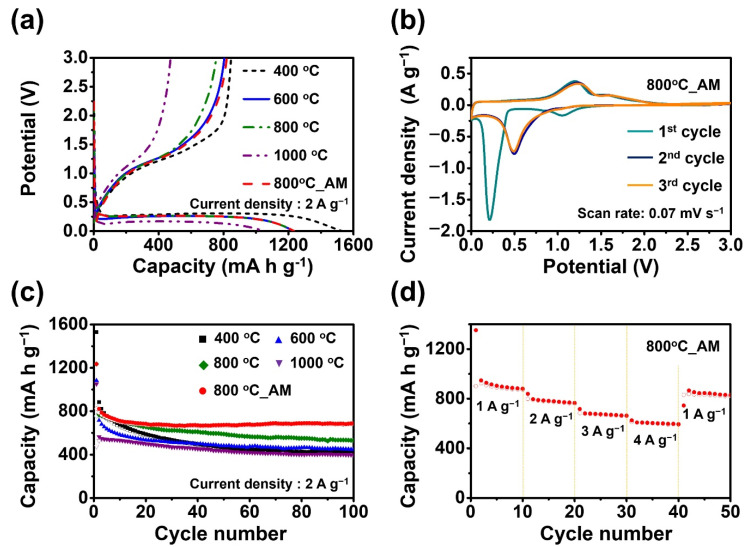
Electrochemical properties of the ZnMn_2_O_4_ powders post-treated at various temperatures and aggregated microsphere consisting of ZnMn_2_O_4_ nanoparticles prepared at 800 °C (800 °C_AM); (**a**) the initial charge and discharge curves, (**b**) cyclic voltammograms of 800 °C_AM, (**c**) cycling performances at a constant current density of 2.0 A g^−1^, (**d**) rate performances of 800 °C_AM.

## Data Availability

The data presented in this study are available on request from the corresponding author.

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
