# Peer review of "Morphological and Electrochemical Properties of ZnMn2O4 Nanopowders and Their Aggregated Microspheres Prepared by Simple Spray Drying Process"

_nanomaterials, 2022, doi:10.3390/nano12040680_

Round 1

Reviewer 1 Report

In this manuscript, ZnMn2O4 nanopowders were prepared by spray drying followed with post-annealing treatment. This article presents the morphology changes during different annealing temperature and the sample fabricated at 800°C exhibits the 687 mAh g-1 after 100th cycles. However, there are some deficiencies, that prevent me from recommending the paper in the current form.
1. In Figure 1. XRD pattern, the standard PDF card of ZnMn2O4 is missing. The author should supply this key parameter.

2. In Figure 3 and Figure 4 EDS mapping images, the author should supply the O element image to confirm the all elements of ZnMn2O4 are uniformly distributed within the whole particle.

3. In Figure 7b electrochemical properties. What’s the accurate electrochemical reaction of this long plateau at about 0.2 V? and during the second cycles, what’s the reaction during Li ions intercalation-extracting process? the author should supply corresponded evidences.

4. The electrochemical properties of ZnMn2O4 is insufficient, the author should supply more evidence to illustrate the electrochemical performances.

5. There are some mistakes within this article, the author should carefully check and correct.

Author Response

Thank you for your helpful comments and revision.

Reviewer #1 In this manuscript, ZnMn2O4 nanopowders were prepared by spray drying followed with post-annealing treatment. This article presents the morphology changes during different annealing temperature and the sample fabricated at 800°C exhibits the 687 mA h g-1 after 100th cycles. However, there are some deficiencies, that prevent me from recommending the paper in the current form.

[Q1] In Figure 1. XRD pattern, the standard PDF card of ZnMn2O4 is missing. The author should supply this key parameter.

[A1] We highly appreciate the reviewer’s positive evaluation on our manuscript. As the reviewer’s comment, we added the standard PDF card of ZnMn2O4 in the revised Figure 1. Therefore, the following data and the relevant sentence was modified in the revised manuscript.

“Figure 1 shows the XRD pattern of the as-prepared and post-treated powders annealed at different temperatures.”

-> “Figure 1 shows the XRD pattern of the as-prepared and post-treated powders annealed at different temperatures and standard PDF card of ZnMn2O4 [27].”

 [27] Min, X.; Zhang, Y.; Yu, M.; Wang, Y.; Yuan, A.; Xu, J. A hierarchical dual-carbon supported ZnMn2O4/C composite as an anode material for Li-ion batteries, J. Alloys Compd. 2021, 877, 160242.

[Q2] In Figure 3 and Figure 4 EDS mapping images, the author should supply the O element image to confirm the all elements of ZnMn2O4 are uniformly distributed within the whole particle.

[A2] As the reviewer’s comment, we added O element image to confirm all elements of ZnMn2O4. Therefore, the following data and relevant sentences were added and modified in the revised manuscript.

“The corresponding dot-mapping images (Figure 3d and 4d) indicated that the zinc and manganese components were uniformly distributed all over the ZnMn2O4 nanopowders.”

->“The corresponding dot-mapping images (Figure 3d and 4d) indicated that the zinc, manganese, and oxygen components were uniformly distributed all over the ZnMn2O4 nanopowders.”

[Q3] In Figure 7b electrochemical properties. What’s the accurate electrochemical reaction of this long plateau at about 0.2 V? and during the second cycles, what’s the reaction during Li ions intercalation-extracting process? the author should supply corresponded evidences.

[A3] In recent, the various ZnMn2O4 anode materials were reported due to their excellent electrochemical properties and their electrochemical reaction mechanism was specifically investigated by in-situ or ex-situ XRD and XANES analysis. Islam et al. performed a structural evolution study during lithiation/delithiation using in-situ XRD and ex-situ synchrotron X-ray absorption spectroscopy techniques to gain an insight into the discharge-charge mechanism of ZnMn2O4 [31]. By their specific analysis, it was confirmed that the final discharge product was a mixture of Zn0, LiZn alloy, and MnO with a fraction of Mn0 embedded in Li2O. Therefore, the electrochemical reaction mechanism is as follows.

The initial discharge

ZnMn2O4 + (x+2z)Li+ + (x+2z)e- → xLiZn + (1-x)Zn + yMn + (2-y)MnO + zLi2O

The reversible reaction

LiZn + Zn + Mn + MnO + (4+t)Li2O ↔ 2ZnO + 2MnO + 9Li + tLi2O

[31] Islam, M.; Ali, G.; Akbar, M.; Ali, B.; Jeong, M.-G.; Kim, J.-Y.; Chung, K. Y.; Nam, K.-W.; Jung, H.-G. Investigating the energy storage performance of the ZnMn2O4 anode for its potential application in lithium-ion batteries, Int. J. Energy Res. 2021, DOI: 10.1002/er.7581.

Therefore, the following sentences were modified in the revised manuscript.

“In the initial discharge curves, all the ZnMn2O4 powders showed one plateau at about 0.3 V, which could be attributed to the reduction of ZnMn2O4 into metallic manganese and zinc.”

->“In the initial discharge curves, all the ZnMn2O4 powders showed one plateau at about 0.3 V, which could be attributed to the reduction of ZnMn2O4 into LiZn, metallic Zn, Mn, MnO, and Li2O as confirmed by previous literature [31].”

“An intensive reduction peak at about 0.2 V was observed in the first cycle, which could be attributed to the irreversible reduction of ZnMn2O4 with concomitant crystal structure destruction to form metallic nanograins (Zn0, Mn0) dispersed in an amorphous Li2O matrix.”

->“An intensive reduction peak at about 0.2 V was observed in the first cycle, which could be attributed to the irreversible reduction of ZnMn2O4 with concomitant crystal structure destruction to form metallic nanograins (Zn0, Mn0) dispersed in an amorphous Li2O matrix and some of LiZn alloys and MnO phases [11,31,32].”

“In recent, the various ZnMn2O4 anode materials were reported due to their excellent electrochemical properties and their electrochemical reaction mechanism was specifically investigated by in-situ or ex-situ XRD and XANES analysis. By specific analysis, it was confirmed that the final discharge product was a mixture of Zn0, LiZn alloy, and MnO with a fraction of Mn0 embedded in Li2O. After second cycle, the reversible reaction is as follows: LiZn + Zn + Mn + MnO + (4+x)Li2O ↔ 2ZnO + 2MnO +9Li + xLi2O [31].”

[Q4] The electrochemical properties of ZnMn2O4 is insufficient, the author should supply more evidence to illustrate the electrochemical performances.

[A4] As the reviewer’s comment, to supply more evidence to illustrate the electrochemical performances, we additionally conducted electrochemical impedance spectra (EIS) analysis of ZnMn2O4 powders post-treated at 800oC temperatures and aggregated microsphere consisting of ZnMn2O4 nanoparticles (800oC_AM). In addition, the morphological properties of ZnMn2O4 powders post-treated at 800oC temperatures and aggregated microsphere consisting of ZnMn2O4 nanoparticles (800oC_AM) after 100 cycles were compared to show structural merits of ZnMn2O4 800oC_AM electrode.

Therefore, the following data and the relevant sentences were added in the revised manuscript.

“Electrochemical impedance spectra (EIS) of ZnMn2O4 powders post-treated at 800oC temperatures and aggregated microsphere consisting of ZnMn2O4 nanoparticles (800oC_AM) obtained before and after 100 cycles are displayed in Figure S5. Before cycle and after 100 cycels, the ZnMn2O4 nanopowders post-treated at 800°C and exhibited aggregated microsphere consisting of ZnMn2O4 nanoparticles (800oC_AM) showed similar Rct (charge-transfer resistance) value. On the other hand, before cycle, ZnMn2O4 nanopowders post-treated at 800oC temperatures showed the low slope (s, Warburg impedance coefficient) of the real part of the impedance spectra (Zre) versus w-1/2 indicating a high lithium-ion diffusion rate of nanopowders comparing to aggregated microsphere. However, after 100 cycles, the slopes of two electrodes showed similar value revealing that the ZnMn2O4 nanopowder became more agglomerated as the cycle progresses. It was confirmed by the SEM images of after 100 cycles as displayed in Figure S6. The ZnMn2O4 nanopowder showed severely aggregated state after 100 cycles. On the other hand, ZnMn2O4 800oC_AM microsphere maintained their spherical and porous structure even after repeated cycles.”

[Q5] There are some mistakes within this article, the author should carefully check and correct.

[A5] As the reviewer’s comment, we carefully modified the overall manuscript.

“Transition metal oxides with various compositions are considered to be promising anode materials for lithium ion batteries (LIBs) due to their high theoretical capacities.”

-> “Transition metal oxides (TMO) with high theoretical capacities have been actively researched as prospective anode materials for lithium ion batteries (LIBs).”

“Lately, several studies have reported the electrochemical properties of ZnMn2O4 powders with various structures, prepared by liquid solution methods, such as sol-gel, co-precipitation, hydrothermal, and solvothermal processes.”

-> “Lately, several studies have reported the electrochemical properties of ZnMn2O4 powders with various structures, prepared by liquid solution methods.”

“Deng et al. have established a single-source precursor route for the preparation of agglomerated phase-pure spinel ZnMn2O4 nanoparticles,”

-> “Deng et al. reported the synthesis of agglomerated ZnMn2O4 nanoparticles by a single-source precursor route,”

“Spray drying is one of the most preferred commercial processes for the preparation of dry powders from a liquid solution or slurry. Thus far, several reports have demonstrated the use of spray drying for the preparation of nanoporous spherical materials, for applications in lithium ion batteries.”

-> “Spray drying is considered as commercial process to obtain dried powders. In general, liquid or colloidal solution was applied to spray drying process resulting in spherical dried powders. Thus far, a number of literatures have been reported to synthesis of electrode materials for lithium ion batteries.”

“In this study, ZnMn2O4 nanopowders with hollow structure were prepared by using a commercial spray drying system followed by post-annealing at different temperatures (Figure S1). The precursor spray solutions were obtained by dissolving pre-determined quantities of zinc nitrate hexahydrate [Zn(NO3)2·6H2O] and manganese nitrate hexahydrate [Mn(NO3)2·6H2O] in distilled water. The total concentration of zinc and manganese components was fixed at 0.5 M. Similarly, the concentration of the chelating agent citric acid was kept at 0.5 M. The temperature at the inlet and outlet of the spray dryer was 350°C and 150°C, respectively. A two-fluid nozzle was used as an atomizer, and the atomization pressure was 0.3 bar. The ZnMn2O4 nanopowders were obtained by spray drying the precursor solution, which were subsequently post-annealed in air at temperatures between 400 and 1000°C for 3 h.”

-> “In this study, ZnMn2O4 nanopowders with hollow structure were obtained by applying a spray drying process followed by post-treatment at different temperatures (Figure S1). The precursor spray solutions were prepared by dissolving zinc nitrate hexahydrate [Zn(NO3)2·6H2O] and manganese nitrate hexahydrate [Mn(NO3)2·6H2O] in distilled water. The concentrations of zinc and manganese components were fixed at 0.17 and 0.33 M, respectively. Similarly, the concentration of the chelating agent citric acid was kept at 0.5 M. The inlet and outlet temperatures during spray drying were controlled as 350°C and 150°C, respectively. The atomizer was applied by a two-fluid. The flow rate and nozzle inlet diameter (where solution is supplied into hot chamber) were 1ml s-1 and 1.0 mm, respectively. Atomization pressure was 0.3 bar. The ZnMn2O4 nanopowders were obtained by spray drying the precursor solution, which were subsequently post-annealed in air at temperatures between 400 and 1000°C for 3 h.”

“After ultrasonication for 1 h by colloidal solution consisting of ZnMn2O4 nanopowders, which were prepared at 800oC post-treatment temperature, the colloidal solution was used to prepare aggregated ZnMn2O4 microsphere powders in the second spray-drying step. The inlet and outlet temperatures of the spray dryer were 300 and 120oC, respectively, and the atomization pressure of the two-fluid nozzle was 2 bar.”

->“To obtain uniformly dispersed state of colloidal solution consisting of ZnMn2O4 nanopowders, which were obtained at 800oC post-treatment temperature, ultrasonication for 1 h was conducted. Subsequently, aggregated ZnMn2O4 microsphere powders were produced by second spray drying process. During the second spray drying, the temperatures of inlet and outlet were 300 and 120oC, respectively. Atomization pressure was higher value (2 bar) comparing to the first spray drying.”

“The mean crystallite size of the ZnMn2O4 powders post-annealed at 400, 600, 800, and 1000°C were estimated to be 10, 20, 32, and 42 nm, respectively, as calculated from the half-width of the (211) peak using Scherrer’s equation.”

-> “The mean crystallite size of the ZnMn2O4 powders post-annealed at 400, 600, 800, and 1000°C were estimated to be 10, 20, 32, and 42 nm, respectively. It was calculated by Scherrer’s equation from the half-width of the (211) peak.”

“The clear lattice fringes separated by lattice spacing of 0.49 nm were seen from the high-resolution TEM image. This lattice spacing value corresponded to the (101) plane of the ZnMn2O4.”

-> “The HR TEM image revealed lattice fringes divided by 0.49 nm, corresponding to (101) plane of the ZnMn2O4.”

“Figure 6 shows the SEM images of the aggregated microsphere consisting of ZnMn2O4 nanoparticles, which were prepared at 800oC post-treatment temperature, formed by the second spray drying process. The nanopowders shown in Figure 4a and 4b were prepared by simple milling process via hand using an agate mortal to obtain the colloidal spray solution for the second step of spray drying. The aggregated powders obtained directly by the second spray drying process were spherical and the micron sized powders were non-aggregated. The crystal structure of the aggregated ZnMn2O4 microsphere powders was maintained even after second spray drying process as shown in Figure S2. The N2 adsorption and desorption isotherms and Barrett–Joyner–Halenda (BJH) pore size distributions of the aggregated microsphere powders formed by the second step spray drying process are shown in Figure S3. The aggregated microsphere powders obtained before post-treatment had well-developed mesopores.”

-> “Figure 6 exhibits the morphologies of the aggregated microsphere consisting of ZnMn2O4 nanoparticles, which were prepared at 800oC post-treatment temperature, formed by the 2 step spray drying process. The nanoparticles consisting of aggregated microsphere were prepared by facile hand-milling process using an agate mortar as shown in Figure 4a and 4b. The colloidal spray solution having ZnMn2O4 nanoparticles was applied to second step of spray drying. In particular, the aggregated microsphere showed non-aggregated state between microsphere and spherical morphology, which could be advantageous for electrode materials.    The crystal structure of the aggregated ZnMn2O4 microsphere powders was kept even after second spray drying process as shown in Figure S2. The surface area and pore size distribution of the aggregated microsphere powders formed by the second step spray drying process are exhibited in Figure S3. The aggregated ZnMn2O4 microsphere powders had well-developed mesopores.”

“The irreversible capacity loss of the ZnMn2O4 nanopowders in the first cycle can be attributed to the formation of a solid electrolyte interphase film and the decomposition of the electrolyte.”

-> “In the initial cycle, ZnMn2O4 electrodes showed commonly irreversible capacity loss, which was owing to the generation of a solid electrolyte interphase (SEI) film.”

“The current density increased from 1.0 A g-1 to 4.0 A g-1 in a stepwise manner and then returned to 1.0 A g-1. The reversible discharge capacities of the ZnMn2O4 nanopowders decreased from 880 to 594 mA h g-1 as the current density increased from 1.0 to 4.0 A g-1.”

->“The rate performance was conducted by increasing current density from 1.0 to 4.0 A g-1 and finally recovered to 1.0 A g-1. Then, ZnMn2O4 nanopowders exhibited discharge capacities of from 880 to 594 mA h g-1 as increasing current density from 1.0 to 4.0 A g-1.

Reviewer 2 Report

Recommendation: major revision.

Comments: In this work, the authors reported a series of ZnMn2O4 nanopowders as anode for lithium-ion batteries via a simple spray drying process using zinc and manganese salts as precursors, citric acid as a chelating agent, followed by post-annealing at different temperatures. The experimental is reasonably designed and the electrochemical performance is good. The experiment data relevant to the interconnected ZnMn2O4 nanopowders offered in this manuscript are sufficient to support the conclusion. But, the structure evolution and storage mechanism of these ZnMn2O4 nanopowders need more data to support. Therefore, I recommend that this manuscript can be accepted after major revision.

  1. The charge/discharge profiles at different current density better offer in the manuscript.
  2. The XRD or TEM characterization of ZnMn2O4nanopowders after cycles may add to investigate the structural stability this anode undergoes during the charge/discharge process.
  3. TheCV analysis for the charge/discharge mechanism on page 7 “form metallic nanograins (Zn0, Mn0 )” maybe not be Some reports reveal that the Zn0 can further react with Li to form LiZn.
  4. To demonstrate the good electrochemical performance, the full-cell performance is better to offer in the manuscript.
  5. The authors should compare the electrochemical performance with reported ZnMn2O4-based anode materials.
  6. There are some writing mistakes in the manuscript. The authors should carefully check and correct them.

Author Response

Answers to the Reviewer’s comments

Thank you for your helpful comments and revision.

Reviewer #2: In this work, the authors reported a series of ZnMn2O4 nanopowders as anode for lithium-ion batteries via a simple spray drying process using zinc and manganese salts as precursors, citric acid as a chelating agent, followed by post-annealing at different temperatures. The experimental is reasonably designed and the electrochemical performance is good. The experiment data relevant to the interconnected ZnMn2O4 nanopowders offered in this manuscript are sufficient to support the conclusion. But, the structure evolution and storage mechanism of these ZnMn2O4 nanopowders need more data to support. Therefore, I recommend that this manuscript can be accepted after major revision.

[Q1] The charge/discharge profiles at different current density better offer in the manuscript.

[A1] We highly appreciate the reviewer’s positive evaluation on our manuscript. As the reviewer’s comment, we added the charge and discharge profiles of ZnMn2O4 800oC_AM at different current density from rate performance data of 800oC_AM. Therefore, the following data and the relevant sentences were added in the revised manuscript.

“Figure S4 exhibited the charge and discharge profiles of ZnMn2O4 800oC_AM at different current density. Despite the increase in current density, potential profile shape well maintained indicating their outstanding rate capability.”

[Q2] The XRD or TEM characterization of ZnMn2O4 nanopowders after cycles may add to investigate the structural stability this anode undergoes during the charge/discharge process.

[A2] As the reviewer’s comment, the morphological properties of ZnMn2O4 powders post-treated at 800oC temperatures and aggregated microsphere consisting of ZnMn2O4 nanoparticles (800oC_AM) after 100 cycles were compared to show structural merits of ZnMn2O4 800oC_AM electrode. Please understand that XRD and TEM measurement cannot be performed under our individual situation, so we have replaced it with SEM analysis.

Therefore, the following data and the relevant sentences were added in the revised manuscript.

“It was confirmed by the SEM images of after 100 cycles as displayed in Figure S6. The ZnMn2O4 nanopowder showed severely aggregated state after 100 cycles. On the other hand, ZnMn2O4 800oC_AM microsphere maintained their spherical and porous structure even after repeated cycles.”

[Q3] The CV analysis for the charge/discharge mechanism on page 7 “form metallic nanograins (Zn0, Mn0 )” maybe not be Some reports reveal that the Zncan further react with Li to form LiZn.

[A3] In recent, the various ZnMn2O4 anode materials were reported due to their excellent electrochemical properties and their electrochemical reaction mechanism was specifically investigated by in-situ or ex-situ XRD and XANES analysis. Islam et al. performed a structural evolution study during lithiation/delithiation using in-situ XRD and ex-situ synchrotron X-ray absorption spectroscopy techniques to gain an insight into the discharge-charge mechanism of ZnMn2O4 [31]. By their specific analysis, it was confirmed that the final discharge product was a mixture of Zn0, LiZn alloy, and MnO with a fraction of Mn0 embedded in Li2O. Therefore, the electrochemical reaction mechanism is as follows.

The initial discharge

ZnMn2O4 + (x+2z)Li+ + (x+2z)e- → xLiZn + (1-x)Zn + yMn + (2-y)MnO + zLi2O

The reversible reaction

LiZn + Zn + Mn + MnO + (4+t)Li2O ↔ 2ZnO + 2MnO + 9Li + tLi2O

[31] Islam, M.; Ali, G.; Akbar, M.; Ali, B.; Jeong, M.-G.; Kim, J.-Y.; Chung, K. Y.; Nam, K.-W.; Jung, H.-G. Investigating the energy storage performance of the ZnMn2O4 anode for its potential application in lithium-ion batteries, Int. J. Energy Res. 2021, DOI: 10.1002/er.7581.

Therefore, the following sentences were modified in the revised manuscript.

“In the initial discharge curves, all the ZnMn2O4 powders showed one plateau at about 0.3 V, which could be attributed to the reduction of ZnMn2O4 into metallic manganese and zinc.”

->“In the initial discharge curves, all the ZnMn2O4 powders showed one plateau at about 0.3 V, which could be attributed to the reduction of ZnMn2O4 into LiZn, metallic Zn, Mn, MnO, and Li2O as confirmed by previous literature[31].”

“An intensive reduction peak at about 0.2 V was observed in the first cycle, which could be attributed to the irreversible reduction of ZnMn2O4 with concomitant crystal structure destruction to form metallic nanograins (Zn0, Mn0) dispersed in an amorphous Li2O matrix.”

->“An intensive reduction peak at about 0.2 V was observed in the first cycle, which could be attributed to the irreversible reduction of ZnMn2O4 with concomitant crystal structure destruction to form metallic nanograins (Zn0, Mn0) dispersed in an amorphous Li2O matrix and some of LiZn alloys and MnO phases[11,31,32].”

“In recent, the various ZnMn2O4 anode materials were reported due to their excellent electrochemical properties and their electrochemical reaction mechanism was specifically investigated by in-situ or ex-situ XRD and XANES analysis. By specific analysis, it was confirmed that the final discharge product was a mixture of Zn0, LiZn alloy, and MnO with a fraction of Mn0 embedded in Li2O. After second cycle, the reversible reaction is as follows : LiZn + Zn + Mn + MnO + (4+x)Li2O ↔ 2ZnO + 2MnO +9Li + xLi2O [31].”

[Q4] To demonstrate the good electrochemical performance, the full-cell performance is better to offer in the manuscript.

[A4] As the reviewer’s comment, the full-cell performance is better to demonstrate the good electrochemical performance. In case of full cell test, stable cathode materials for LIBs should be prepared and it is also demanded to acquire optimized cathode/anode full cell system. Unfortunately, we hope that you will understand that the full-cell test in our group is difficult.

In the future, we will strive to conduct full-cell test through the development of various anode materials as well as optimized cathode materials for lithium-ion batteries at the same time.

[Q5] The authors should compare the electrochemical performance with reported ZnMn2O4-based anode materials.

[A5] As the reviewer’s comment, we added a table data comparing the electrochemical performance of other previously reported ZnMn2O4 materials for lithium-ion batteries. The relevant sentences and Table S1 have been newly added in the revised manuscript.

“The electrochemical properties with other ZnMn2O4 materials as anode materials for LIBs reported in the previous literatures is exhibited in Table S1. The structural advantages of ZnMn2O4 800oC_AM facilitated good cycle life with high reversible capacities, even comparable to that of other ZnMn2O4 materials.”

References

[S1] Zhang, G.; Yu, L.; Wu, H. B.; Hoster, H. E.; Lou, X. W. Formation of ZnMn2O4 ball-in-ball hollow microspheres as a high-performance anode for lithium-ion batteries, Adv. Mater. 2012, 24, 4609-4613.

[S2] Song, M. S.; Cho, Y. J.; Yoon, D. Y.; Nahm, S.; Oh, S. H.; Woo, K.; Ko, J. M.; Cho, W. I. Solvothermal synthesis of ZnMn2O4 as an anode materialin lithium ion battery, Electrochim. Acta. 2014, 137, 266-272.

[S3] Wang, N.; Ma, X.; Xu, H.; Chen, L.; Yue, J.; Niu, F.; Yang, J.; Qian, Y. Porous ZnMn2O4 microspheres as a promising anode material for advanced lithium-ion batteries, Nano Energy 2014, 6, 193-199.

[S4] Zhang, L.; Zhu, S.; Cao, H.; Hou, L.; Yuan, C. Hierarchical porous ZnMn2O4 hollow nanotubes with enhanced lithium storage toward lithium-ion batteries, Chem. Eur. J. 2015, 21, 10771-10777.

[S5] Dang, W.; Wang, F.; Ding, Y.; Feng, C.; Guo, Z. Synthesis and electrochemical properties of ZnMn2O4 microspheres for lithium-ion battery application, J. Alloys Compd. 2017, 690, 72-79.

[S6] Chen, X.; Zhang, Y.; Lin, H.; Xia, P.; Cai, X.; Li, X.; Li, X.; Li, W. Porous ZnMn2O4 nanospheres: Facile synthesis through microemulsion method and excellent performance as anode of lithium ion battery, J. Power Sources. 2016, 312, 137-145.

[S7] Zhang, Y.; Zhang, Y.; Guo, C.; Tang, B.; Wang, X.; Bai, Z. Porous ZnMn2O4 nanowires as an advanced anode material for lithium ion battery, Electrochim. Acta 2015, 182, 1140-1144.

[Q6] There are some writing mistakes in the manuscript. The authors should carefully check and correct them.

[A6] As the reviewer’s comment, we carefully modified the overall manuscript.

“Transition metal oxides with various compositions are considered to be promising anode materials for lithium ion batteries (LIBs) due to their high theoretical capacities.”

-> “Transition metal oxides (TMO) with high theoretical capacities have been actively researched as prospective anode materials for lithium ion batteries (LIBs).”

“Lately, several studies have reported the electrochemical properties of ZnMn2O4 powders with various structures, prepared by liquid solution methods, such as sol-gel, co-precipitation, hydrothermal, and solvothermal processes.”

-> “Lately, several studies have reported the electrochemical properties of ZnMn2O4 powders with various structures, prepared by liquid solution methods.”

“Deng et al. have established a single-source precursor route for the preparation of agglomerated phase-pure spinel ZnMn2O4 nanoparticles,”

-> “Deng et al. reported the synthesis of agglomerated ZnMn2O4 nanoparticles by a single-source precursor route,”

“Spray drying is one of the most preferred commercial processes for the preparation of dry powders from a liquid solution or slurry. Thus far, several reports have demonstrated the use of spray drying for the preparation of nanoporous spherical materials, for applications in lithium ion batteries.”

-> “Spray drying is considered as commercial process to obtain dried powders. In general, liquid or colloidal solution was applied to spray drying process resulting in spherical dried powders. Thus far, a number of literatures have been reported to synthesis of electrode materials for lithium ion batteries.”

“In this study, ZnMn2O4 nanopowders with hollow structure were prepared by using a commercial spray drying system followed by post-annealing at different temperatures (Figure S1). The precursor spray solutions were obtained by dissolving pre-determined quantities of zinc nitrate hexahydrate [Zn(NO3)2·6H2O] and manganese nitrate hexahydrate [Mn(NO3)2·6H2O] in distilled water. The total concentration of zinc and manganese components was fixed at 0.5 M. Similarly, the concentration of the chelating agent citric acid was kept at 0.5 M. The temperature at the inlet and outlet of the spray dryer was 350°C and 150°C, respectively. A two-fluid nozzle was used as an atomizer, and the atomization pressure was 0.3 bar. The ZnMn2O4 nanopowders were obtained by spray drying the precursor solution, which were subsequently post-annealed in air at temperatures between 400 and 1000°C for 3 h.”

-> “In this study, ZnMn2O4 nanopowders with hollow structure were obtained by applying a spray drying process followed by post-treatment at different temperatures (Figure S1). The precursor spray solutions were prepared by dissolving zinc nitrate hexahydrate [Zn(NO3)2·6H2O] and manganese nitrate hexahydrate [Mn(NO3)2·6H2O] in distilled water. The concentrations of zinc and manganese components were fixed at 0.17 and 0.33 M, respectively. Similarly, the concentration of the chelating agent citric acid was kept at 0.5 M. The inlet and outlet temperatures during spray drying were controlled as 350°C and 150°C, respectively. The atomizer was applied by a two-fluid. The flow rate and nozzle inlet diameter (where solution is supplied into hot chamber) were 1ml s-1 and 1.0 mm, respectively. Atomization pressure was 0.3 bar. The ZnMn2O4 nanopowders were obtained by spray drying the precursor solution, which were subsequently post-annealed in air at temperatures between 400 and 1000°C for 3 h.”

“After ultrasonication for 1 h by colloidal solution consisting of ZnMn2O4 nanopowders, which were prepared at 800oC post-treatment temperature, the colloidal solution was used to prepare aggregated ZnMn2O4 microsphere powders in the second spray-drying step. The inlet and outlet temperatures of the spray dryer were 300 and 120oC, respectively, and the atomization pressure of the two-fluid nozzle was 2 bar.”

->“To obtain uniformly dispersed state of colloidal solution consisting of ZnMn2O4 nanopowders, which were obtained at 800oC post-treatment temperature, ultrasonication for 1 h was conducted. Subsequently, aggregated ZnMn2O4 microsphere powders were produced by second spray drying process. During the second spray drying, the temperatures of inlet and outlet were 300 and 120oC, respectively. Atomization pressure was higher value (2 bar) comparing to the first spray drying.”

“The mean crystallite size of the ZnMn2O4 powders post-annealed at 400, 600, 800, and 1000°C were estimated to be 10, 20, 32, and 42 nm, respectively, as calculated from the half-width of the (211) peak using Scherrer’s equation.”

-> “The mean crystallite size of the ZnMn2O4 powders post-annealed at 400, 600, 800, and 1000°C were estimated to be 10, 20, 32, and 42 nm, respectively. It was calculated by Scherrer’s equation from the half-width of the (211) peak.”

“The clear lattice fringes separated by lattice spacing of 0.49 nm were seen from the high-resolution TEM image. This lattice spacing value corresponded to the (101) plane of the ZnMn2O4.”

-> “The HR TEM image revealed lattice fringes divided by 0.49 nm, corresponding to (101) plane of the ZnMn2O4.”

“Figure 6 shows the SEM images of the aggregated microsphere consisting of ZnMn2O4 nanoparticles, which were prepared at 800oC post-treatment temperature, formed by the second spray drying process. The nanopowders shown in Figure 4a and 4b were prepared by simple milling process via hand using an agate mortal to obtain the colloidal spray solution for the second step of spray drying. The aggregated powders obtained directly by the second spray drying process were spherical and the micron sized powders were non-aggregated. The crystal structure of the aggregated ZnMn2O4 microsphere powders was maintained even after second spray drying process as shown in Figure S2. The N2 adsorption and desorption isotherms and Barrett–Joyner–Halenda (BJH) pore size distributions of the aggregated microsphere powders formed by the second step spray drying process are shown in Figure S3. The aggregated microsphere powders obtained before post-treatment had well-developed mesopores.”

-> “Figure 6 exhibits the morphologies of the aggregated microsphere consisting of ZnMn2O4 nanoparticles, which were prepared at 800oC post-treatment temperature, formed by the 2 step spray drying process. The nanoparticles consisting of aggregated microsphere were prepared by facile hand-milling process using an agate mortar as shown in Figure 4a and 4b. The colloidal spray solution having ZnMn2O4 nanoparticles was applied to second step of spray drying. In particular, the aggregated microsphere showed non-aggregated state between microsphere and spherical morphology, which could be advantageous for electrode materials.    The crystal structure of the aggregated ZnMn2O4 microsphere powders was kept even after second spray drying process as shown in Figure S2. The surface area and pore size distribution of the aggregated microsphere powders formed by the second step spray drying process are exhibited in Figure S3. The aggregated ZnMn2O4 microsphere powders had well-developed mesopores.”

“The irreversible capacity loss of the ZnMn2O4 nanopowders in the first cycle can be attributed to the formation of a solid electrolyte interphase film and the decomposition of the electrolyte.”

-> “In the initial cycle, ZnMn2O4 electrodes showed commonly irreversible capacity loss, which was owing to the generation of a solid electrolyte interphase (SEI) film.”

“The current density increased from 1.0 A g-1 to 4.0 A g-1 in a stepwise manner and then returned to 1.0 A g-1. The reversible discharge capacities of the ZnMn2O4 nanopowders decreased from 880 to 594 mA h g-1 as the current density increased from 1.0 to 4.0 A g-1.”

->“The rate performance was conducted by increasing current density from 1.0 to 4.0 A g-1 and finally recovered to 1.0 A g-1. Then, ZnMn2O4 nanopowders exhibited discharge capacities of from 880 to 594 mA h g-1 as increasing current density from 1.0 to 4.0 A g-1.

Reviewer 3 Report

The paper presents an interesting study of shaping ZnMn2O4 into microparticles made of nanoparticles using spray-drying, as an anode material for Li-ion batteries. The results are properly presented and show the interest of a two-step spray-drying process. However, the paper lacks accuracy in the result description and comparison to literature that prevents publication.

  • The introduction overlooks recent work on spray-drying from metal salt solutions. I found at least two papers, relevant for Li-ion battery applications: a cathode preparation (A. Hou 2020 J. Mater. Sci. Mat. Electr., 10.1007/s10854-019-02627-9) and an anode preparation (J Asenbauer 2020 ChemSusChem, 10.1002/cssc.202000559).This part must be completed with relevant discussion of the literature.
  • In general, comparison with the literature on ZnMn2O4 anode materials is not up-to-date, with the most recent cited paper published in 2013. A series of good papers have been published on the subject in the past 2-5 years. On top of the other citations in my review, the paper by X. Luo 2018 ACS Appl Mater Interfaces, 10.1021/acsami.8b10111 provides interesting conclusions and literature review. This section of the introduction must be improved.
  • In the Materials and Methods section, the sample preparation does not mention flow rates or nozzle size, so that it would be impossible to reproduce the results. The text tells about a two-fluid nozzle, while the scheme in Figure S1 shows a single solution of metal salt precursor, which is misleading. As for the second spray-drying process, the composition of the precursor suspension of colloidal ZnMn2O4 is not described at all.
  • Figure 1 shows X-ray diffractograms with too thick lines in a too small frame, so that details are hidden. The figure should include a reference XRD pattern of ZnMn2O4 and identify the unattributed peaks. Please refer to the excellent figure 1(a) in the paper by X. Min 2021 J. All. Comp., 10.1016/j.jallcom.2021.160242.
  • In the legends of Fig. 2, 3, 4 and 5, and on page 6 line 166, “mortal” must be corrected to “mortar”.
  • The text explains that ZnMn2O4 reduces to Zn0 and Mn0 at the first cycle. The reader then wonders what kind of material cycles as the anode in the next 99 cycles in Fig7c. Please discuss this point in detail. You can refer to the interesting paper by T. Feng 2021 Trans. Nonferrous Met. Soc. China, 10.1016/S1003-6326(21)65493-6, or any other relevant paper on this subject. Alternatively, post-mortem XRD analysis would be of interest.
  • The blue and green lines down in Fig.7a can not be seen.
  • The conclusion states that the “second spray drying process provides structural stability” to the ZnMn2O4 nanoparticles. I did not see any discussion of this point in the paper. Please provide evidence.

Author Response

Answers to the Reviewer’s comments

Thank you for your helpful comments and revision.

Reviewer #3: The paper presents an interesting study of shaping ZnMn2O4 into microparticles made of nanoparticles using spray-drying, as an anode material for Li-ion batteries. The results are properly presented and show the interest of a two-step spray-drying process. However, the paper lacks accuracy in the result description and comparison to literature that prevents publication:

[Q1] The introduction overlooks recent work on spray-drying from metal salt solutions. I found at least two papers, relevant for Li-ion battery applications: a cathode preparation (A. Hou 2020 J. Mater. Sci. Mat. Electr., 10.1007/s10854-019-02627-9) and an anode preparation (J Asenbauer 2020 ChemSusChem, 10.1002/cssc.202000559). This part must be completed with relevant discussion of the literature.

[A1] We highly appreciate the reviewer’s positive evaluation on our manuscript. As the reviewer’s comment, excellent two papers, which were related to work on spray-drying from metal salt solutions, were additionally discussed and the literatures were cited in the revised manuscript.

Therefore, the following sentences were added and modified and the relevant papers were cited in the revised manuscript.

“Hou et al. reported synthesis of LiNi0.815Co0.15Al0.035O2 cathode materials by spray drying and a high-temperature calcination. The metal salt dissolved solution was applied to preparation of precursor powders [25]. Asenbauer et al. synthesized microsized, nanocrystalline Zn0.9Fe0.1O-C secondary particles by three spray-drying steps. The large secondary particle size of about 10-15 mm facilitated handling and processing during preparation slurry and provided good electrochemical properties [26].”

“However, to the best of our knowledge, preparation of multicomponent oxide nanopowders and their aggregated powders by spray drying of aqueous spray solution containing metal salts has rarely been reported, due to the poor drying characteristics of metal salts under a humid atmosphere.”

->“However, preparation of multicomponent oxide nanopowders and their aggregated powders by spray drying using aqueous spray solution should be researched due to the poor drying characteristics of metal salts under a humid atmosphere.”

[25] Hou, A.; Liu, Y.; Ma, L.; Chai, F.; Zhang, P.; Fan, Y. High-rate LiNi0.815Co0.15Al0.035O2 cathode material prepared by spray drying method for Li-ion batteries. J. Mater. Sci. Mater. Electronics 2020, 31, 1159-1167.

[26] Asenbauer, J.; Binder, J. R.; Mueller, F.; Kuenzel, M.; Geiger, D.; Kaiser, U.; Passerini, S.; Bresser, D. Scalable synthesis of microsized, nanocrystalline Zn0.9Fe0.1O-C secondary particles and their use in Zn0.9Fe0.1O-C/LiNi0.5Mn1.5O4 lithium-ion full cells, ChemSusChem 2020,13, 3504-3513.

[Q2] In general, comparison with the literature on ZnMn2O4 anode materials is not up-to-date, with the most recent cited paper published in 2013. A series of good papers have been published on the subject in the past 2-5 years. On top of the other citations in my review, the paper by X. Luo 2018 ACS Appl Mater Interfaces, 10.1021/acsami.8b10111 provides interesting conclusions and literature review. This section of the introduction must be improved.

[A2] As the reviewer’s comment, the excellent relevant paper were additionally discussed and the literatures were cited in the revised manuscript.

Therefore, the following sentences were added and modified and the relevant papers were cited in the revised manuscript.

“Luo et al. synthesized mesoporous ZnMn2O4 microtyblues via a facile biomorphic strategy employing biotemplate [16]. The 1D-mesoporous structure of ZnMn2O4 microtyblues facilitated superior electrochemical properties of high capacity and rate capability.”

[16] Luo, X.; Zhang, X.; Chen, L.; Li, L.; Zhu, G.; Chen, G.; Yan, D.; Xu, H.; Yu. A. Mesoporous ZnMn2O4 microtubules derived from a biomorphic strategy for high-performance lithium/sodium ion batteries, ACS Appl. Mater. Interfaces 2018, 10, 33170-33178.

[Q3] In the Materials and Methods section, the sample preparation does not mention flow rates or nozzle size, so that it would be impossible to reproduce the results. The text tells about a two-fluid nozzle, while the scheme in Figure S1 shows a single solution of metal salt precursor, which is misleading. As for the second spray-drying process, the composition of the precursor suspension of colloidal ZnMn2O4 is not described at all.

[A3] As the reviewer’s comment, we added additional information of flow rates and nozzle size in the materials and methods section. Moreover, schematic illustrations of spray drying system and formation of aggregated ZnMn2O4 microsphere were modified and added in the revised manuscript.

Therefore, the following data and the relevant sentences were modified in the revised manuscript.

“The flow rate and nozzle inlet diameter (where solution is supplied into hot chamber) were 1ml s-1 and 1.0 mm, respectively.”

“Figure S1 showed schematic diagram of spray drying system and formation of ZnMn2O4 aggregated microsphere by 2 step spray drying process.”

[Q4] Figure 1 shows X-ray diffractograms with too thick lines in a too small frame, so that details are hidden. The figure should include a reference XRD pattern of ZnMn2O4 and identify the unattributed peaks. Please refer to the excellent figure 1(a) in the paper by X. Min 2021 J. All. Comp., 10.1016/j.jallcom.2021.160242.

[A4] As the reviewer’s comment, we added the standard PDF card of ZnMn2O4 in the revised Figure 1. Therefore, the following data and the relevant sentences were modified in the revised manuscript.

“Figure 1 shows the XRD pattern of the as-prepared and post-treated powders annealed at different temperatures.”

-> “Figure 1 shows the XRD pattern of the as-prepared and post-treated powders annealed at different temperatures and standard PDF card of ZnMn2O4.”

In addition, the relevant paper was cited in the revised manuscript.

[27] Min, X.; Zhang, Y.; Yu, M.; Wang, Y.; Yuan, A.; Xu, J. A hierarchical dual-carbon supported ZnMn2O4/C composite as an anode material for Li-ion batteries, J. Alloys Compd. 2021, 877, 160242.

[Q5] In the legends of Fig. 2, 3, 4 and 5, and on page 6 line 166, “mortal” must be corrected to “mortar”.

[A5] As the reviewer’s comment, “mortal” in the legends of Fig. 2, 3, 4 and 5, and on page 6 line 166 was modified to “mortar”

[Q6] The text explains that ZnMn2O4 reduces to Zn0 and Mn0 at the first cycle. The reader then wonders what kind of material cycles as the anode in the next 99 cycles in Fig7c. Please discuss this point in detail. You can refer to the interesting paper by T. Feng 2021 Trans. Nonferrous Met. Soc. China, 10.1016/S1003-6326(21)65493-6, or any other relevant paper on this subject. Alternatively, post-mortem XRD analysis would be of interest.

[A6] In recent, the various ZnMn2O4 anode materials were reported due to their excellent electrochemical properties and their electrochemical reaction mechanism was specifically investigated by in-situ or ex-situ XRD and XANES analysis. Islam et al. performed a structural evolution study during lithiation/delithiation using in-situ XRD and ex-situ synchrotron X-ray absorption spectroscopy techniques to gain an insight into the discharge-charge mechanism of ZnMn2O4 [31]. By their specific analysis, it was confirmed that the final discharge product was a mixture of Zn0, LiZn alloy, and MnO with a fraction of Mn0 embedded in Li2O. Therefore, the electrochemical reaction mechanism is as follows.

The initial discharge

ZnMn2O4 + (x+2z)Li+ + (x+2z)e- → xLiZn + (1-x)Zn + yMn + (2-y)MnO + zLi2O

The reversible reaction

LiZn + Zn + Mn + MnO + (4+t)Li2O ↔ 2ZnO + 2MnO + 9Li + tLi2O

[31] Islam, M.; Ali, G.; Akbar, M.; Ali, B.; Jeong, M.-G.; Kim, J.-Y.; Chung, K. Y.; Nam, K.-W.; Jung, H.-G. Investigating the energy storage performance of the ZnMn2O4 anode for its potential application in lithium-ion batteries, Int. J. Energy Res. 2021, DOI: 10.1002/er.7581.

Therefore, the following sentences were modified in the revised manuscript.

“In the initial discharge curves, all the ZnMn2O4 powders showed one plateau at about 0.3 V, which could be attributed to the reduction of ZnMn2O4 into metallic manganese and zinc.”

->“In the initial discharge curves, all the ZnMn2O4 powders showed one plateau at about 0.3 V, which could be attributed to the reduction of ZnMn2O4 into LiZn, metallic Zn, Mn, MnO, and Li2O as confirmed by previous literature [31].”

“An intensive reduction peak at about 0.2 V was observed in the first cycle, which could be attributed to the irreversible reduction of ZnMn2O4 with concomitant crystal structure destruction to form metallic nanograins (Zn0, Mn0) dispersed in an amorphous Li2O matrix.”

->“An intensive reduction peak at about 0.2 V was observed in the first cycle, which could be attributed to the irreversible reduction of ZnMn2O4 with concomitant crystal structure destruction to form metallic nanograins (Zn0, Mn0) dispersed in an amorphous Li2O matrix and some of LiZn alloys and MnO phases[11,31,32].”

“In recent, the various ZnMn2O4 anode materials were reported due to their excellent electrochemical properties and their electrochemical reaction mechanism was specifically investigated by in-situ or ex-situ XRD and XANES analysis. By specific analysis, it was confirmed that the final discharge product was a mixture of Zn0, LiZn alloy, and MnO with a fraction of Mn0 embedded in Li2O. After second cycle, the reversible reaction is as follows : LiZn + Zn + Mn + MnO + (4+x)Li2O ↔ 2ZnO + 2MnO +9Li + xLi2O [31].”

In addition, the relevant paper was cited in the revised manuscript.

[32] Feng, T.-T.; Yang, J.; Dai, S.-Y.; Wang, J.-C.; Wu, M.-Q. Microemulsion synthesis of ZnMn2O4/Mn3O4 sub-microrods for Li-ion batteries and their conversion reaction mechanism, Trans. Nonferrous Met. Soc. China 2021, 31, 265−276.

[Q7] The blue and green lines down in Fig.7a can not be seen.

[A7] As the reviewer’s comment, we modified the Figure 7a to clearly show the blue and green lines.

[Q8] The conclusion states that the “second spray drying process provides structural stability” to the ZnMn2O4 nanoparticles. I did not see any discussion of this point in the paper. Please provide evidence.

[A8] As the reviewer’s comment, the morphological properties of ZnMn2O4 powders post-treated at 800oC temperatures and aggregated microsphere consisting of ZnMn2O4 nanoparticles (800oC_AM) after 100 cycles were compared to show structural merits of ZnMn2O4 800oC_AM electrode.

Therefore, the following data and the relevant sentences were added in the revised manuscript.

“It was confirmed by the SEM images of after 100 cycles as displayed in Figure S6 The ZnMn2O4 nanopowder showed severely aggregated state after 100 cycles. On the other hand, ZnMn2O4 800oC_AM microsphere maintained their spherical and porous structure even after repeated cycles.”

Round 2

Reviewer 1 Report

The authors have fully addressed all my questions, and the manuscript can be accepted now.

Reviewer 2 Report

Accept in present form

Reviewer 3 Report

The paper has been improved nicely.

Some spell check is required (examples: 

Lines 48 & 49: microtyblues to be corrected in microtubules

Line 192: In recent to be changed for Recently

Line 226: cycels to be corrected in cycles).